# Learning From Small Samples: An Analysis of Simple Decision Heuristics

**Özgür Şimşek** and **Marcus Buckmann**
Center for Adaptive Behavior and Cognition
Max Planck Institute for Human Development
Lentzeallee 94, 14195 Berlin, Germany
{ozgur, buckmann}@mpib-berlin.mpg.de

## Abstract

Simple decision heuristics are models of human and animal behavior that use few pieces of information—perhaps only a single piece of information—and integrate the pieces in simple ways, for example, by considering them sequentially, one at a time, or by giving them equal weight. We focus on three families of heuristics: single-cue decision making, lexicographic decision making, and tallying. It is unknown how quickly these heuristics can be learned from experience. We show, analytically and empirically, that substantial progress in learning can be made with just a few training samples. When training samples are very few, tallying performs substantially better than the alternative methods tested. Our empirical analysis is the most extensive to date, employing 63 natural data sets on diverse subjects.

## 1 Introduction

You may remember that, on January 15, 2009, in New York City, a commercial passenger plane struck a flock of geese within two minutes of taking off from LaGuardia Airport. The plane immediately and completely lost thrust from both engines, leaving the crew facing a number of critical decisions, one of which was whether they could safely return to LaGuardia. The answer depended on many factors, including the weight, velocity, and altitude of the aircraft, as well as wind speed and direction. None of these factors, however, are directly involved in how pilots make such decisions. As copilot Jeffrey Skiles discussed in a later interview [1], pilots instead use a single piece of visual information: whether the desired destination is staying stationary in the windshield. If the destination is rising or descending, the plane will undershoot or overshoot the destination, respectively. Using this visual cue, the flight crew concluded that LaGuardia was out of reach, deciding instead to land on the Hudson River. Skiles reported that subsequent simulation experiments consistently showed that the plane would indeed have crashed before reaching the airport.

Simple decision heuristics, such as the one employed by the flight crew, can provide effective solutions to complex problems [2, 3]. Some of these heuristics use a single piece of information; others use multiple pieces of information but combine them in simple ways, for example, by considering them sequentially, one at a time, or by giving them equal weight.

Our work is concerned with two questions: How effective are simple decision heuristics? And how quickly can they be learned from experience? We focus on problems of comparison, where the objective is to decide which of a given set of objects has the highest value on an unobserved criterion. These problems are of fundamental importance in intelligent behavior. Humans and animals spend much of their time choosing an object to act on, with respect to some criterion whose value is unobserved at the time. Choosing a mate, a prey to chase, an investment strategy for a retirement fund, or a publisher for a book are just a few examples. Earlier studies on this problem have shown

that simple heuristics are surprisingly accurate in natural environments [4, 5, 6, 7, 8, 9], especially when learning from small samples [10, 11].

We present analytical and empirical results on three families of heuristics: lexicographic decision making, tallying, and single-cue decision making. Our empirical analysis is the most extensive to date, employing 63 natural environments on diverse subjects. Our main contributions are as follows: (1) We present analytical results on the rate of learning heuristics from experience. (2) We show that very few learning instances can yield effective heuristics. (3) We empirically investigate single-cue decision making and find that its performance is remarkable. (4) We find that the most robust decision heuristic for small sample sizes is tallying. Collectively, our results have important implications for developing more successful heuristics and for studying how well simple heuristics capture human and animal decision making.

## 2  Background

The comparison problem asks which of a given set of objects has the highest value on an unobserved criterion, given a number of attributes of the objects. We focus on *pairwise comparisons*, where exactly two objects are being compared. We consider a decision to be accurate if it selects the object with the higher criterion value (or either object if they are equal in criterion value). In the heuristics literature, *attributes* are called *cues*; we will follow this custom when discussing heuristics.

The heuristics we consider decide by comparing the objects on one or more cues, asking which object has the higher cue value. Importantly, they do not require the difference in cue value to be quantified. For example, if we use *height of a person* as a cue, we need to be able to determine which of two people is taller but we do not need to know the height of either person or the magnitude of the difference. Each cue is associated with a direction of inference, also known as *cue direction*, which can be positive or negative, favoring the object with the higher or lower cue value, respectively. Cue directions (and other components of heuristics) can be learned in a number of ways, including social learning. In our analysis, we learn them from training examples.

*Single-cue* decision making is perhaps the simplest decision method one can imagine. It compares the objects on a single cue, breaking ties randomly. We learn the identity of the cue and its direction from a training sample. Among the $2k$ possible models, where $k$ is the number of cues, we choose the $\langle$cue, direction$\rangle$ combination that has the highest accuracy in the training sample, breaking ties randomly.

*Lexicographic heuristics* consider the cues one at a time, in a specified order, until they find a cue that *discriminates* between the objects, that is, one whose value differs on the two objects. The heuristic then decides based on that cue alone. An example is take-the-best [12], which orders cues with respect to decreasing *validity* on the training sample, where validity is the accuracy of the cue among pairwise comparisons on which the cue discriminates between the objects.

*Tallying* is a voting model. It determines how each cue votes on its own (selecting one or the other object or abstaining from voting) and selects the object with the highest number of votes, breaking ties randomly. We set cue directions to the direction with highest validity in the training set.

Paired comparison can also be formulated as a classification problem. Let $y_A$ denote the criterion value of object A, $\mathbf{x}_A$ the vector of attribute values of object A, and $\Delta y_{AB} = y_A - y_B$ the difference in criterion values of objects A and B. We can define the *class f* of a pair of objects as a function of the difference in their criterion values:

$$f(\Delta y_{AB}) \;\; = \;\; \begin{cases} 1 & \text{if } \Delta y_{AB} > 0 \\ -1 & \text{if } \Delta y_{AB} < 0 \\ 0 & \text{if } \Delta y_{AB} = 0 \end{cases}$$

A class value of $1$ denotes that object A has the higher criterion value, $-1$ that object B has the higher criterion value, and $0$ that the objects are equal in criterion value. The comparison problem is intrinsically symmetrical: comparing A to B should give us the same decision as comparing B to A. That is, $f(\Delta y_{AB})$ should equal $-f(\Delta y_{BA})$. Because the latter equals $-f(-\Delta y_{AB})$, we have the following symmetry constraint: $f(z) = -f(-z)$, for all $z$. We can expect better classification accuracy if we impose this symmetry constraint on our classifier.

# 3 Building blocks of decision heuristics

We first examine two building blocks of learning heuristics from experience: assigning cue direction and determining which of two cues has the higher predictive accuracy. The former is important for all three families of heuristics whereas the latter is important for lexicographic heuristics when determining which cue should be placed first. Both components are building blocks of heuristics in a broader sense—their use is not limited to the three families of heuristics considered here.

Let $A$ and $B$ be the objects being compared, $x_A$ and $x_B$ denote their cue values, $y_A$ and $y_B$ denote their criterion values, and $sgn$ denote the mathematical sign function: $sgn(x)$ is 1 if $x > 0$, 0 if $x = 0$, and $-1$ if $x < 0$. A single training instance is the tuple $\langle sgn(x_A - x_B), \ sgn(y_A - y_B) \rangle$, corresponding to a single pairwise comparison, indicating whether the cue and the criterion change from one object to the other, along with the direction of change. For example, if $x_A = 1$, $y_A = 10$, $x_B = 2$, $y_B = 5$, the training instance is $\langle -1, +1 \rangle$.

**Learning cue direction.** We assume, without loss of generality, that cue direction in the population is positive (we ignore the case where the cue direction in the population is neutral). Let $p$ denote the success rate of the cue in the population, where *success* is the event that the cue decides correctly. We examine two probabilities, $e_1$ and $e_2$. The former is the probability of correctly inferring the cue direction from a set of training instances. The latter is the probability of deciding correctly on a new (unseen) instance using the direction inferred from the training instances.

We define an *informative* instance to be one in which the objects differ both in their cue values and in their criterion values, a *positive* instance to be one in which the cue and the criterion change in the same direction ($\langle 1, 1 \rangle$ or $\langle -1, -1 \rangle$), and a *negative* instance to be one in which the cue and the criterion change in the opposite direction ($\langle 1, -1 \rangle$ or $\langle -1, 1 \rangle$).

Let $n$ be the number of training instances, $n_+$ the number of positive training instances, and $n_-$ the number of negative training instances. Our estimate of cue direction is positive if $n_+ > n_-$, negative if $n_+ < n_-$, and a random choice between positive and negative if $n_+ = n_-$.

Given a set of independent, informative training instances, $n_+$ follows the binomial distribution with $n$ trials and success probability $p$, allowing us to write $e_1$ as follows:

$$
\begin{aligned}
e_1 &= P(n_+ > n_-) + \frac{1}{2} P(n_+ = n_-) \\
&= \sum_{k=\lfloor n/2 \rfloor + 1}^{n} \binom{n}{k} p^k (1-p)^{n-k} + I(n \text{ is even}) \frac{1}{2} \binom{n}{n/2} p^{n/2}(1-p)^{n/2},
\end{aligned}
$$

where $I$ is the indicator function. After one training instance, $e_1$ equals $p$. After one more instance, $e_1$ remains the same. This is a general property: After an odd number of training instances, an additional instance does not increase the probability of inferring the direction correctly.

On a new (test) instance, the cue decides correctly with probability $p$ if cue direction is inferred correctly and with probability $1 - p$ otherwise. Consequently, $e_2 = pe_1 + (1-p)(1-e_1)$.

Simple algebra yields the following expected learning rates: After $2k + 1$ training instances, with two additional instances, the increase in the probability of inferring cue direction correctly is $(2p - 1)(p(1-p))^{k+1}$ and the increase in the probability of deciding correctly is $(2p - 1)^2(p(1-p))^{k+1}$ .

Figure 1 shows $e_1$ and $e_2$ as a function of training-set size $n$ and success rate $p$. The more predictive the cue is, the smaller the sample needs to be for a desired level of accuracy in both $e_1$ and $e_2$. This is of course a desirable property: The more useful the cue is, the faster we learn how to use it correctly.

The figure also shows that there are highly diminishing returns, from one odd training-set size to the next, as the size of the training set increases. In fact, just a few instances make great progress toward the maximum possible. The third plot in the figure reveals this property more clearly. It shows $e_2$ divided by its maximum possible value ($p$) showing how quickly we reach the maximum possible accuracy for cues of various predictive ability. The minimum value depicted in this figure is 0.83, observed at $n = 1$. This means that even after a single training instance, our expected accuracy is at least 83% of the maximum accuracy we can reach. And this value rises quickly with each additional pair of training instances.

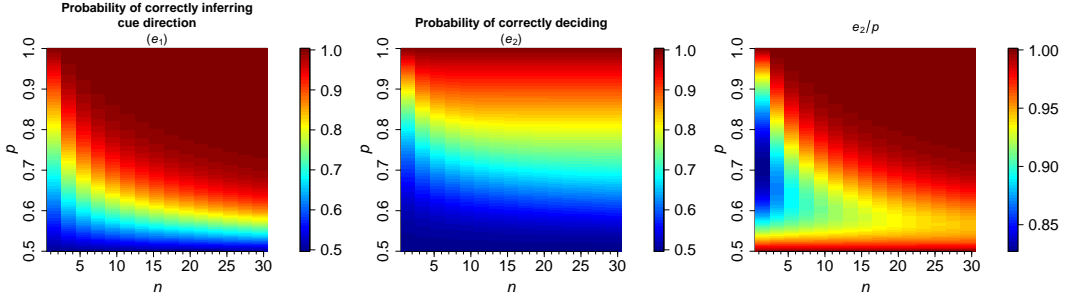

Figure 1: Learning cue direction.

**Learning to order two cues.** Assume we have two cues with success rates $p$ and $q$ in the population, with $p > q$. We expand the definition of informative instance to require that the objects differ on the second cue as well. We examine two probabilities, $e_3$ and $e_4$. The former is the probability of ordering the two cues correctly, which means placing the cue with higher success rate above the other one. The latter is the probability of deciding correctly with the inferred order. We chose to examine learning to order cues independently of learning cue directions. One reason is that people do not necessarily learn the cue directions from experience. In many cases, they can guess the cue direction correctly through causal reasoning, social learning, past experience in similar problems, or other means. In the analysis below, we assume that the directions are assigned correctly.

Let $s_1$ and $s_2$ be the success rates of the two cues in the training set. If instances are informative and independent, $s_1$ and $s_2$ follow the binomial distribution with parameters $(n, p)$ and $(n, q)$, allowing us to write $e_3$ as follows:

$$e_3 = P(s_1 > s_2) + \frac{1}{2}P(s_1 = s_2) = \sum_{0 \leq j < i \leq n} P(s_1 = i)P(s_2 = j) + \frac{1}{2}\sum_{i=0}^{n} P(s_1 = i)P(s_2 = i)$$

After one training instance, $e_3$ is $0.5 + 0.5(p-q)$, which is a linear function of the difference between the two success rates.

If we order cues correctly, a decision on a test instance is correct with probability $p$, otherwise with probability $q$. Thus, $e_4 = pe_3 + q(1 - e_3)$.

Figure 2 shows $e_3$ and $e_4$ as a function of $p$ and $q$ after three training instances. In general, larger values of $p$, as well as larger differences between $p$ and $q$, require smaller training sets for a desired level of accuracy. In other words, learning progresses faster where it is more useful. The third plot in the figure shows $e_4$ relative to the maximum value it can take, the maximum of $p$ and $q$. The minimum value depicted in this figure is 90.9%. If we examine the same figure after only a single training instance, we see that this minimum value is 86.6% (figure not shown).

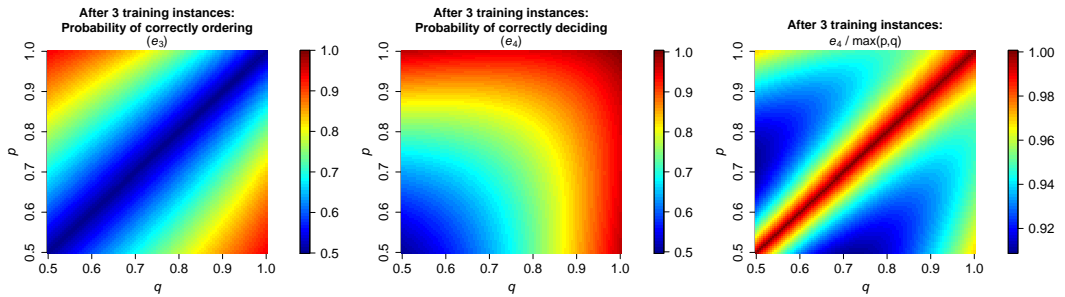

Figure 2: Learning cue order.

# 4 Empirical analysis

We next present an empirical analysis of 63 natural data sets, most from two earlier studies [4, 13]. Our primary objective is to examine the empirical learning rates of heuristics. From the analytical results of the preceding section, we expect learning to progress rapidly. A secondary objective is to examine the effectiveness of different ways cues can be ordered in a lexicographic heuristic.

The data sets were gathered from a wide variety of sources, including online data repositories, textbooks, packages for R statistical software, statistics and data mining competitions, research publications, and individual scientists collecting field data. The subjects were diverse, including biology, business, computer science, ecology, economics, education, engineering, environmental science, medicine, political science, psychology, sociology, sports, and transportation. The data sets varied in size, ranging from 13 to 601 objects. Many of the smaller data sets contained the entirety of the population of objects, for example, all 29 islands in the Galápagos archipelago. The data sets are described in detail in the supplementary material.

We present results on lexicographic heuristics, tallying, single-cue decision making, logistic regression, and decision trees trained by CART [14]. We used the CART implementation in rpart [15] with the default splitting criterion Gini, cp=0, minsplit=2, minbucket=1, and 10-fold cross-validated cost-complexity pruning. There is no explicit way to implement the symmetry constraint for decision trees; we simply augmented the training set with its mirror image with respect to the direction of comparison. For logistic regression, we used the glm function of R, setting the intercept to zero to implement the symmetry constraint. To the glm function, we input the cues in the order of decreasing correlation with the criterion so that the weakest cues were dropped first when the number of training instances was smaller than the number of cues.

**Ordering cues in lexicographic heuristics.** We first examine the different ways lexicographic heuristics can order the cues. With $k$ cues, there are $k!$ possible cue orders. Combined with the possibility of using each cue with a positive or negative direction, there are $2^k k!$ possible lexicographic models, a number that increases very rapidly with $k$. How should we choose one if our top criterion is accuracy but we also want to pay attention to computational cost and memory requirements?

We consider three methods. The first is a greedy search, where we start by deciding on the first cue to be used (along with its direction), then the second, and so on, until we have a fully specified lexicographic model. When deciding on the first cue, we select the one that has the highest validity in the training examples. When deciding on the $m$th cue, $m \geq 2$, we select the cue that has the highest validity in the examples left over after using the first $m - 1$ cues, that is, those examples where the first $m - 1$ cues did not discriminate between the two objects. The second method is to order cues with respect to their validity in the training examples, as take-the-best does. Evaluating cues independently of each other substantially reduces computational and memory requirements but perhaps at the expense of accuracy. The third method is to use the lexicographic model—among the $2^k k!$ possibilities—that gives the highest accuracy in the training examples. Identifying this rule is NP-complete [16, 17], and it is unlikely to generalize well, but it will be informative to examine it. The three methods have been compared earlier [18] on a data set consisting of German cities [12], where the fitting accuracy of the best, greedy, validity, and random ordering was 0.758, 0.756, 0.742, and 0.700, respectively.

Figure 3 (top panel) shows the fitting accuracy of each method in each of the 63 data sets when all possible pairwise comparisons were conducted among all objects. Because of the long simulation time required, we show an approximation of the best ordering in data sets with seven or more cues. In these data sets, we started with the two lexicographic rules generated by the greedy and the validity ordering, kept intact the cues that were placed seventh or later in the sequence, and tested all possible permutations of their first six cues, trying out both possible cue directions. The figure also shows the mean accuracy of random ordering, where cues were used in the direction of higher validity. In all data sets, greedy ordering was identical or very close in accuracy to the best ordering. In addition, validity ordering was very close to greedy ordering except in a handful of data sets. One explanation is that a continuous cue that is placed first in a lexicographic model makes all (or almost all) decisions and therefore the order of the remaining cues does not matter. We therefore also examine the binary version of each data set where numerical cues were dichotomizing around the median (Figure 3 bottom panel). There was little difference in the relative positions of greedy and optimal ordering except in one data set. There was more of a drop in the relative accuracy of

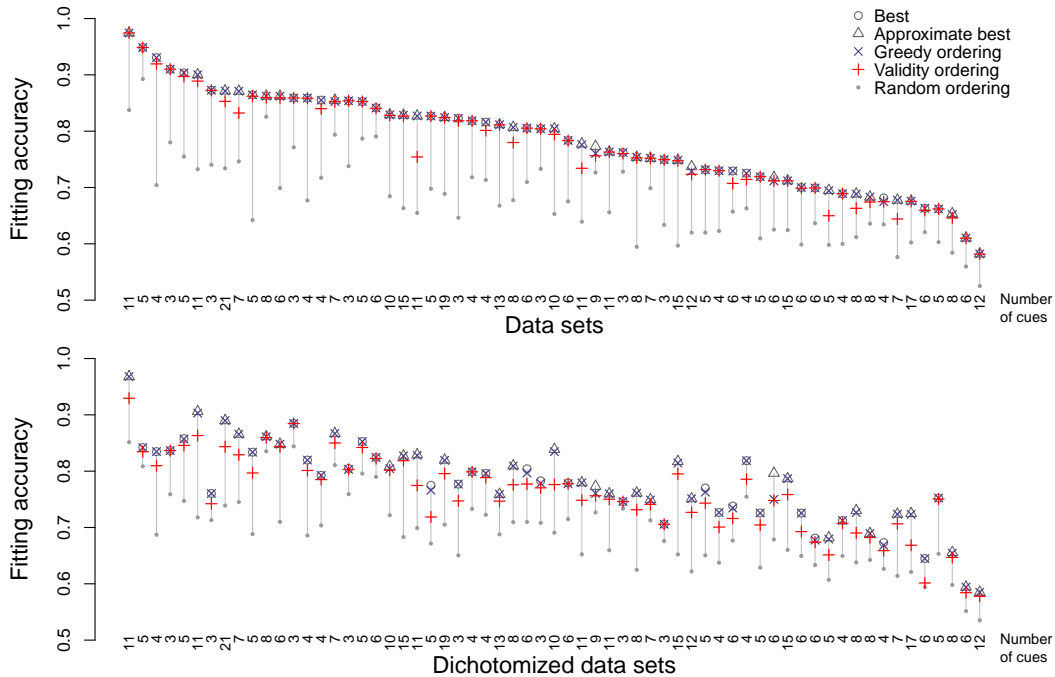

Figure 3: Fitting accuracy of lexicographic models, with and without dichotomizing the cues.

the validity ordering, but this method still achieved accuracy close to that of the best ordering in the majority of the data sets.

We next examine predictive accuracy. Figure 4 shows accuracies when the models were trained on 50% of the objects and tested on the remaining 50%, conducting all possible pairwise comparisons within each group. Mean accuracy across data sets was 0.747 for logistic regression, 0.746 for CART, 0.743 for greedy lexicographic and take-the-best, 0.734 for single-cue, and 0.725 for tallying. Figure 5 shows learning curves, where we grew the training set one pairwise comparison at a time. Two individual objects provided a single instance for training or testing and were never used again, neither in training nor in testing. Consequently, the training instances were independent of each other but they were not always informative (as defined in Section 3). The figure shows the mean learning curve across all data sets as well as individual learning curves on 16 data sets. We present the graphs without error bars for legibility; the highest standard error of the data points displayed is 0.0014 in Figure 4 and 0.0026 in Figure 5.

A few observations are noteworthy: (1) Heuristics were indeed learned rapidly. (2) In the early part of the learning curve, tallying generally had the highest accuracy. (3) The performance of single-cue was remarkable. When trained on 50% of the objects, its mean performance was better than tallying, 0.9 percentage points behind take-the-best, and 1.3 percentage points behind logistic regression. (4) Take-the-best performed better than or as well as greedy lexicographic in most data sets. A detailed comparison of the two methods is provided below.

**Validity versus greedy ordering in lexicographic decision making.** The learning curves on individual data sets took one of four forms: (1) There was no difference in any part of the learning curve. This is the case when a continuous cue is placed first: This cue almost always discriminates between the objects, and cues further down in the sequence are seldom (if ever) used. Because greedy and validity ordering always agree on the first cue, the learning curves are identical or nearly so. Twenty-two data sets were in this first category. (2) Validity ordering was better than greedy ordering in some parts of the learning curve and never worse. This category included 35 data sets. (3) Learning curves crossed: Validity ordering generally started with higher accuracy than greedy ordering; the difference diminished with increasing training-set size, and eventually greedy ordering exceeded validity ordering in accuracy (2 data sets). (4) Greedy ordering was better than validity or-

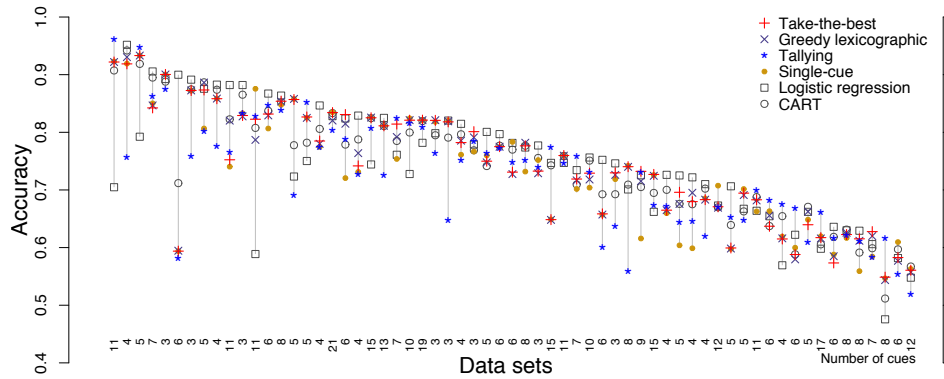

Figure 4: Predictive accuracy when models are trained with 50% of the objects in each data set and tested on the remaining 50%.

dering in some parts of the learning curve and never worse (4 data sets). To draw these conclusions, we considered a difference to be present if the error bars ($\pm$ 2 SE) did not overlap.

## 5  Discussion

We isolated two building blocks of decision heuristics and showed analytically that they require very few training instances to learn under conditions that matter the most: when they add value to the ultimate predictive ability of the heuristic. Our empirical analysis confirmed that heuristics typically make substantial progress early in learning.

Among the algorithms we considered, the most robust method for very small training sets is tallying. Earlier work [11] concluded that take-the-best (with undichotomized cues) is the most robust model for training sets with 3 to 10 objects but tallying (with undichotomized cues) was absent from this earlier study. In addition, we found that the performance of single-cue decision making is truly remarkable. This heuristic has been analyzed [19] by assuming that the cues and the criterion follow the normal distribution; we are not aware of an earlier analysis of its empirical performance on natural data sets.

Our analysis of learning curves differs from earlier studies. Most earlier studies [20, 10, 21, 11, 22] examined performance as a function of number of objects in the training set, where training instances are all possible pairwise comparisons among those objects. Others increased the training set one pairwise comparison at a time but did not keep the pairwise comparisons independent of each other [23]. In contrast, we increased the training set one pairwise comparison at a time and kept all pairwise comparisons independent of each other. This makes it possible to examine the incremental value of each training instance.

There is criticism of decision heuristics because of their computational requirements. For instance, it has been argued that take-the-best can be described as a simple algorithm but its successful execution relies on a large amount of precomputation [24] and that the computation of cue validity in the German city task "would require 30,627 pairwise comparisons just to establish the cue validity hierarchy for predicting city size" [25]. Our results clearly show that the actual computational needs of heuristics can be very low if independent pairwise comparisons are used for training. A similar result—that just a few samples may suffice—exists within the context of Bayesian inference [26].

#### Acknowledgments

Thanks to Gerd Gigerenzer, Konstantinos Katsikopoulos, Malte Lichtenberg, Laura Martignon, Perke Jacobs, and the ABC Research Group for their comments on earlier drafts of this article. This work was supported by Grant SI 1732/1-1 to Özgür Şimşek from the Deutsche Forschungsgemeinschaft (DFG) as part of the priority program "New Frameworks of Rationality" (SPP 1516).

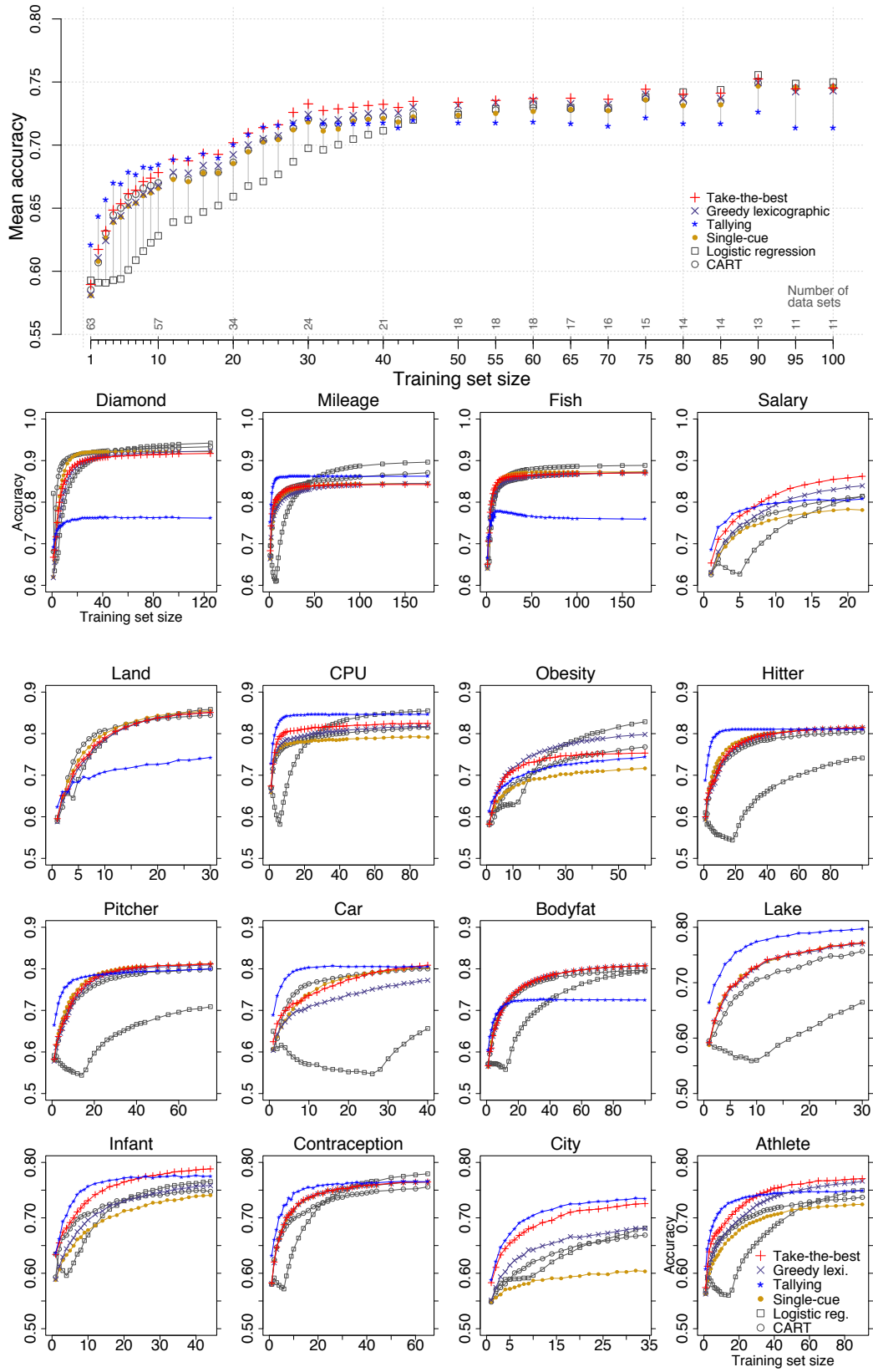

Figure 5: Learning curves.

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
