[Supplementary Material · supplement.pdf]

# Learning From Small Samples: An Analysis of Simple Decision Heuristics

## SUPPLEMENTARY MATERIAL

**Özgür Şimşek** and **Marcus Buckmann**
Center for Adaptive Behavior and Cognition
Max Planck Institute for Human Development
Lentzeallee 94, 14195 Berlin, Germany
{ozgur, buckmann}@mpib-berlin.mpg.de

### Abstract

This document describes the data sets used in the empirical analysis. There were 63 data sets in total. They were obtained from a wide variety of sources, including online data repositories, data mining competitions, textbooks, research publications, packages for R statistical software, and individual scientists collecting field data. The subjects were diverse, including biology, business, computer science, ecology, economics, education, engineering, environmental science, medicine, political science, psychology, sociology, sports, and transportation. The data sets varied in size, ranging from 13 to 601 objects. Number of attributes ranged from 3 to 21.

**AFL** OBJECTS: 41 Australian Football League (AFL) games at the Melbourne Cricket Ground in 1993 and 1994. CRITERION: attendance. ATTRIBUTES: forecasted maximum temperature on the day of the game, total attendance at other AFL games in Melbourne and Geelong on the day of the game, total membership in the two clubs whose teams were playing, number of players in the top 50 who participated in the game, number of days since the earliest game of the season. SOURCE: This data set was assembled by Rowan Todd and Mark McNaughton for a class project at the University of Queensland in a statistics course taught by Margaret Mackisack. The data sources were *The Football Bible '94* by Rex Hunt, *The Weekend Australian*, *Inside Football*, and *Football Record*. The data set is available from OzDASL data library [1], where it is listed with the name *AFL Crowd Attendance at the MCG*.

**Agriculture** OBJECTS: 29 groups of tropical subsistence cultivators. CRITERION: agricultural intensity, defined as the proportion of time that each crop-fallow cycle is in the cropping phase. ATTRIBUTES: population density, whether the group produces livestock, mean annual precipitation, length of dry season, soil type (normal, alluvial/hydromorphic), major staples of the group (root crops, cereal crops). SOURCE: The data set is assembled by Turner, Hanham, and Portararo [2] from earlier publications. It is available electronically from an online repository maintained by Winner [3], where it is listed with the name *population and other factors relating to agricultural intensity*.

**Air** OBJECTS: 41 cities in the United States. CRITERION: annual mean concentration of sulfur dioxide. ATTRIBUTES: average annual temperature, number of manufacturing enterprises employing 20 or more workers, population, average annual wind speed, average annual rainfall, average number of days with rainfall per year. SOURCE: The data were gathered by Sokal and Rohlf [4] from several publications of the United States government. The data set is reported in a book by Hand, Daly, Lunn, McConway, and Ostrowski [5] with identifying number 26 and label *air pollution in US cities*.

**Algae** OBJECTS: 340 samples from European rivers taken over a period of approximately one year. CRITERION: density of algae type a. ATTRIBUTES: concentrations of eight chemicals, season (fall, winter, spring, summer), river size (small, medium, large), fluid velocity (low, medium, high). SOURCE: The data set is from the 1999 Computational Intelligence and Learning (COIL) competition. It is available from the UCI data repository [6], where it is labeled *COIL 1999 competition data*.

**Athlete** OBJECTS: 202 nationally-ranked athletes in Australia. CRITERION: blood hemoglobin concentration. ATTRIBUTES: body mass index, sum of skin folds, percent body fat, lean body mass, height, weight, sex, the sport the athlete competes in (basketball, field, gymnastics, netball, rowing, track 400m, swimming, sprint, tennis, water polo). SOURCE: The data were collected by Telford and Cunningham [7] at the Australian Institute of Sport. The data set is reported by Maindonald and Braun [8] and is available from associated R package *DAAG* [9] with label *ais*.

**Basketball** OBJECTS: 96 basketball players. CRITERION: points scored per minute. ATTRIBUTES: assists per minute, height, time played, age. SOURCE: The data set is reported by Simonoff [10] and is available from a website maintained by the author [11], where it is labeled *baskball.dat*.

**Birthweight** OBJECTS: 189 newborns. CRITERION: birth weight. ATTRIBUTES: age of mother, weight of mother at last menstrual period, race (white, black, other), number of previous premature labors, number of physician visits during the first trimester, presence of uterine irritability, whether the mother smoked during pregnancy, whether the mother has a history of hypertension. SOURCE: The data were collected at Baystate Medical Center in Springfield, Massachusetts in 1986 [12]. The data set is electronically available from R package *MASS* [13, 14], where it is labeled *birthwt*.

**Bodyfat** OBJECTS: 252 males. CRITERION: percentage of body fat determined by underwater weighing. ATTRIBUTES: age, weight, height, and various body circumference measurements: neck, chest, abdomen, hip, thigh, knee, ankle, biceps, forearm, wrist. SOURCE: The data were collected by Penrose, Nelson, and Fisher [15]. The data set is available from StatLib [16] with label *bodyfat*.

**Bone** OBJECTS: 42 male skeletons buried in coffins. CRITERION: nitrogen content. ATTRIBUTES: deposition time, depth of burial, age of the person, whether quicklime was added to the coffin at burial, whether skeleton was contaminated with oil, burial site (2 sites 130 km apart in northern England). SOURCE: The data were collected by Jarvis [17]. The data set is available electronically from a data repository maintained by Winner [3], where it is listed with the name *nitrogen levels in skeletal bones of various ages and internment lengths*.

**Car** OBJECTS: 93 passenger cars on sale in the United States in 1993. CRITERION: sale price of the most basic version of the car. ATTRIBUTES: city mileage, highway mileage, cylinders (3, 4, 5, 6, 8, rotary), engine size, maximum horsepower, engine revolutions per mile in highest gear, fuel tank capacity, passenger capacity, length, wheelbase, width, weight, rear seat room, luggage capacity, u-turn space, airbag (none, driver only, both driver and passenger), whether a manual transmission version is available, whether the manufacturer is from the United States, type of car (small, sporty, compact, midsize, large, van), drive-train type (rear, front, four-wheel drive). SOURCE: The data set was assembled by Lock [18] using information from *PACE New Car & Truck 1993 Buying Guide* and *Consumer Reports April 1993 Annual Auto Issue*. It is available from R package *MASS* [13, 14] with label *Cars93*.

**Cigarette** OBJECTS: 25 brands of cigarettes. CRITERION: carbon monoxide emitted from the cigarette smoke. ATTRIBUTES: weight, tar content, nicotine content. SOURCE: The data were produced by the Federal Trade Commission. The data set is reported by Mendenhall and Sincich [19]. It is electronically available from the *Journal of Statistics Education* [20].

**City** OBJECTS: 76 cities in Germany with more than 100,000 inhabitants. CRITERION: population. ATTRIBUTES: whether the city has a team in the major soccer league *Bundesliga*, whether the city is a state capital, whether the city was formerly in East Germany, whether the city is in the industrial belt, whether the abbreviation for the city on license plates is one-letter long, whether the city is on the intercity train line, whether the city hosted a trade fair in 2013, whether the city is the national capital, whether the city has a university. SOURCE: This data set originally appeared in Gigerenzer and Goldstein [21]. For the current study, it was updated to reflect 2013 population data and attributes. Population data were obtained from the Federal Statistical Office (Das Statistische Bundesamt) based on the 2011 census and population density of revision 31.12.2012. Data on trade

fairs were obtained from AUMA, the Association of the German Trade Fair Industry. The data set reflects only the trade fairs in AUMA category *international and national events*. The industrial belt is the region of Germany known as *Ruhrgebiet*. The intercity train line includes IC and ICE stops. Universities include *Universität*, *Institut für Technologie*, *Technische Universität*, and *Technische Hochschule*.

**Contraception** OBJECTS: 210 localities in the world (most are United Nations members but includes areas like Hong Kong that are not independent countries). CRITERION: percentage of unmarried women using a modern method of contraception. ATTRIBUTES: annual population growth rate, per capita 2001 gross domestic product, percentage of females over the age of 15 who are economically active, population, expected number of live births per female in 2000, percentage of population that is urban in 2001. SOURCE: The data set is reported by Weisberg [22] who notes that the source of the data is the United Nations. It is electronically available from R package *alr3* [23] where it is labeled *UN3*.

**CPU** OBJECTS: 209 central processing units on the market in 1981–1984. CRITERION: published performance on a benchmark mix relative to an IBM 370/158 Model 3. ATTRIBUTES: cycle time, minimum main memory, maximum main memory, cache memory, minimum number of channels, maximum number of channels. SOURCE: The data set was assembled by Ein-Dor and Feldmesser [24] using information from *Computerworld* magazine. It is electronically available from R package *MASS* [13, 14] with label *cpus*.

**Crime** OBJECTS: 47 states of the United States. CRITERION: crime rate in 1960. ATTRIBUTES: percentage of males aged 14–24 in state population, indicator variable for a southern state, mean years of schooling of the population aged 25 years or older, per capita expenditure on police protection in 1960, per capita expenditure on police protection in 1959, labor force participation rate of civilian urban males in the age-group 14–24, number of males per 100 females, state population in 1960, percentage of nonwhites in the population, unemployment rate of urban males 14–24, unemployment rate of urban males 35–39, wealth (median value of transferable assets or family income), income inequality (percentage of families earning below half the median income), probability of imprisonment (ratio of number of commitments to number of offenses), average time served by offenders in state prisons before their first release. SOURCE: The data set was assembled by Ehrlich [25] from various publications of the United States government, including *Uniform Crime Reports* of the Federal Bureau of Investigation, United States Census, and *National Prison Statistics Bulletin*. Rounded data taken from Vandaele [26] is electronically available from OzDASL [1], where it is labeled *uscrime*.

**Detroit** OBJECTS: 13 years (1961–1973) in Detroit, Michigan. CRITERION: number of homicides per 100,000 of population. ATTRIBUTES: average hourly earnings, average weekly earnings, number of government workers, number of manufacturing workers, number of non-manufacturing workers, number of government workers, number of white males, unemployment, percentage of homicides cleared by arrests, per 100,000 population: full-time police, number of handgun licenses, number of handgun registrations. SOURCE: The data set was assembled by Fisher [27]. It appears in books by Miller [28] and Gunst and Mason [29]. It is electronically available from StatLib [16].

**Diamond** OBJECTS: 308 round diamond stones. CRITERION: sale price. ATTRIBUTES: weight in carats, color purity (D, E, F, G, H, I), clarity (internally flawless, very very slight inclusion 1, very very slight inclusion 2, very slight inclusion 1, very slight inclusion 2), certification (Gemmological Institute of America, International Gemmological Institute, Hoge Raad Voor Diamant). SOURCE: The data set was assembled by Chu [30] from advertisements in Singapore's *Business Times* edition of February 18, 2000. It is available electronically from R package *Ecdat* [31].

**Dropout** OBJECTS: 63 public high schools in Chicago. CRITERION: dropout rate. ATTRIBUTES: enrollment, attendance rate, parental involvement rate, percent limited-English students, percent low-income students, average class size, percent White students, percent Black students, percent Hispanic students, percent Asian students, percent minority teachers, average composite ACT score, IGAP scores: reading, math, science, social science, writing. SOURCE: This prediction problem is from a study by Czerlinski, Gigerenzer, and Goldstein [32]. Their data sources are two articles in the February 1995 issue of *Chicago* magazine [33, 34], where the authors note that their primary data source is Illinois State Board of Education's 1994 School Report Card.

**Excavator**  OBJECTS: 33 hydraulic excavators operating in the opencast mining industry in the United Kingdom. CRITERION: annual maintenance cost. ATTRIBUTES: weight, type of machine (front shovel, backacter), type of industry (opencast coal, opencast slate), company attitude to used oil analysis (regular use, not). SOURCE: The data are from a study by Edwards, Holt, and Harris [35]. The data set is available electronically from a data repository maintained by Winner [3], where it is listed with the name *construction plant maintenance costs*.

**Fair**  OBJECTS: 601 married individuals. CRITERION: number of extramarital affairs in the past year. ATTRIBUTES: sex, age, number of years in marriage, whether the individual has children, degree of religiosity (on a scale from 1 to 5), years of education, occupation (on a scale from 1 to 7, according to Hollingshead classification), marital happiness (on a scale from 1 to 5). SOURCE: The data set was assembled by Fair [36] using responses to a survey by *Psychology Today* in 1969. It is reported by Greene [37] and is electronically available from R package *Ecdat* [31].

**Fire**  OBJECTS: 517 forest fires in Montesinho Natural Park, Portugal. CRITERION: burned area. ATTRIBUTES: temperature, relative humidity, wind speed, accumulated precipitation within the previous 30 minutes, month of the year, day of the week, location of the fire: *x*-axis and *y*-axis spatial coordinates within the Montesinho park map, ranging from 1 to 9. SOURCE: The data set was assembled by Cortez and Morais [38] from measurements obtained by Bragança Polytechnic Institute and by the inspector responsible for the Montesinho fire occurrences from January 2000 to December 2003. It is available from the UCI Machine Learning Repository [6].

**Fish**  OBJECTS: 413 female Arctic charr. CRITERION: number of eggs. ATTRIBUTES: age, weight, mean egg weight. SOURCE: This prediction problem is from a study by Czerlinski et al. [32]. The data were collected by Christian Gillet from the French National Institute for Agricultural Research. The data set used in this study was obtained via personal communication in April 2012.

**Fuel**  OBJECTS: 51 states and the District of Columbia of the United States. CRITERION: per capita motor fuel consumption in 2001. ATTRIBUTES: population, fuel tax rate, per capita income, miles of federal-aid primary highways, proportion of the population who are licensed drivers. SOURCE: The data set is reported by Weisberg [22] who notes that the source of the data is the Federal Highway Administration. The data set is available from R package *alr3* [23] where it is labeled *Fuel2001*.

**Galápagos**  OBJECTS: 29 islands in the Galápagos archipelago. CRITERION: number of plant species. ATTRIBUTES: surface area, elevation, distance to the nearest island, surface area of the nearest island, distance from the center of the archipelago. SOURCE: The data set was assembled by Johnson and Raven [39]. It is reported by Weisberg [22] and is available electronically from associated R package *alr3* [23]. Elevations of six very small islands were not recorded in the original data set. These were taken from a version of the data set available from OzDASL [1], labeled *Galápagos Island Species Data*, and where missing elevations were obtained from web searches and large-scale maps.

**Gambling**  OBJECTS: 47 British teenagers. CRITERION: annual gambling expenditure. ATTRIBUTES: sex, socio-economic status, weekly income, verbal score. SOURCE: The data were collected by Ide-Smith and Lea [40]. The data set is reported by Faraway [41] and is electronically available from associated R package *faraway* [42], where it is labeled *teengamb*.

**Highway**  OBJECTS: 39 segments of highway in Minnesota. CRITERION: accident rate. ATTRIBUTES: segment length, average daily traffic count, truck volume as a percent of total volume, speed limit, number of lanes, lane width, shoulder width, number of signalized interchanges per mile, number of freeway-type interchanges per mile, number of access points per mile, highway type (federal interstate highway, principal arterial highway, major arterial, other). SOURCE: The data set is reported by Weisberg [22] who notes that the data were taken from an unpublished master's paper in civil engineering by Carl Hoffstedt. The data set is available electronically from R package *alr3* [23].

**Hitter**  OBJECTS: 322 hitters in North American Major League Baseball. CRITERION: annual salary at the beginning of the 1987 season. ATTRIBUTES: 1986 performance: number of at bats, hits, home runs, runs scored, runs batted in, walks, putouts, assists, errors; career performance: number of at bats, hits, home runs, runs scores, runs batted in, walks; number of years in the major leagues; division at the end of the 1986 season (East, West); league at the end of the 1986 season (American, National); league at the beginning of the 1987 season (American, National). SOURCE:

The data set was prepared by the Statistical Graphics Section of the American Statistical Association for the 1988 Annual Statistical Meetings and is available from StatLib [16]. The version used in this work is from Fox [43], includes corrections by Hoaglin and Velleman [44], and is electronically available from a website maintained by Fox [45].

**Homeless** OBJECTS: 50 cities in the United States. CRITERION: rate of homelessness. AT-TRIBUTES: mean temperature, unemployment rate, percentage of inhabitants with incomes below the poverty line, vacancy rate, population, percentage of public housing, whether the city has rent control. SOURCE: The data set was assembled by Tucker [46] from Department of Housing and Urban Development's 1984 *Report to the Secretary on the Homeless and Emergency Shelters* and other sources.

**House** OBJECTS: 27 houses sold in Erie, Pennsylvania. CRITERION: Selling price. ATTRIBUTES: Current tax, number of bathrooms, number of rooms, number of bedrooms, number of fireplaces, number of garage spaces, lot size, total living space, age of house, construction type (brick, frame, brick and frame, aluminum and frame), style (ranch, two story, one and a half story). SOURCE: The data set is reported by Weisberg [47], who notes that the data are from an article by Narula and Wellington [48]. Partial data set (all attributes except for construction type and style) is electronically available from StatLib [16], in collection *alr*, where it is labeled *alr241*.

**Ice** OBJECTS: 30 four-week periods. CRITERION: ice cream consumption per capita. AT-TRIBUTES: price of ice cream per pint, weekly family income, mean temperature. SOURCE: The data set is reported by Hand et al. [5] with identifying number 268 and name *ice cream consumption*. Their source is an article by Kadiyala [49], who reports that the data are from a study by Hildreth and Lu [50].

**Infant** OBJECTS: 105 nations. CRITERION: infant-mortality rate. ATTRIBUTES: per-capita income, geographic location (Africa, Americas, Asia, Europe), whether the country exports oil. SOURCE: Rates of infant mortality were obtained by Leinhardt and Wasserman [51] from the editorial section of the *New York Times* [52]. The data set is reported by Fox [43] and is electronically available from a website maintained by the author [53].

**Jet** OBJECTS: 22 jet fighter aircraft of the United States Navy and Air Force. CRITERION: First flight date, in months after January 1940. ATTRIBUTES: Specific power, flight range factor, payload as a fraction of gross weight, sustained load factor, whether the aircraft can land on a carrier. SOURCE: The data set was assembled by Stanley and Miller [54]. It is reported in a book by Hand et al. [5] with identifying number 110.

**Lake** OBJECTS: 69 world lakes. CRITERION: number of known crustacean zooplankton species present. ATTRIBUTES: surface area, maximum depth, mean depth, specific conductance, elevation, latitude, longitude, distance to nearest lake, number of lakes within 20 km, rate of photosynthesis. SOURCE: The data set is reported by Weisberg [22] who notes that the data were provided by S. Dodson and discussed in part in Dodson [55]. The data set is electronically available from R package *alr3* [23].

**Land** OBJECTS: 67 counties in Minnesota. CRITERION: rent per acre paid in 1977 for agricultural land planted in alfalfa. ATTRIBUTES: average rent for all tillable land, density of dairy cows, proportion of pasture land, whether liming is required to grow alfalfa. SOURCE: The data set is reported by Weisberg [22] who notes that the data were collected by Douglas Tiffany. The data set is electronically available from R package *alr3* [23] where it is labeled *landrent*.

**Mammal** OBJECTS: 62 mammal species. CRITERION: average daily sleep. ATTRIBUTES: body weight, brain weight, maximum life span, gestation time, predation index, sleep exposure index, overall danger index.. SOURCE: The data are from a study by Allison and Cicchetti [56]. The data set is available from StatLib [16], where it is labeled *sleep*.

**Manpower** OBJECTS: 17 naval hospitals of the United States around the world. CRITERION: monthly man-hours. ATTRIBUTES: average daily patient load, monthly X-ray exposures, monthly occupied bed days, eligible population in the area, average length of stay by a patient. SOURCE: The data were obtained by Myers [57] from a publication of the United States Navy [58]. The data set is available electronically from R package *genridge* [59]. A similar data set is reported in a book by Hand et al. [5] with identifying number 269 and label *hospital data*.

**Men** OBJECTS: 34 famous men. CRITERION: mean attractiveness rating. ATTRIBUTES: mean likeability rating, name recognition, whether the man is American. SOURCE: This prediction problem is from a study by Czerlinski et al. [32]. The data were collected by Henss [60] with the participation of 115 male and 131 female Germans, in ages ranging from 17 to 66 years old.

**Mileage** OBJECTS: 398 cars built in 1970–1982. CRITERION: mileage. ATTRIBUTES: number of cylinders, engine displacement, horsepower, vehicle weight, time to accelerate from 0 to 60 mph, model year, origin (American, European, Japanese). SOURCE: The data set was prepared by the Committee on Statistical Graphics of the American Statistical Association for its Second Exposition of Statistical Graphics Technology, held in conjunction with the Annual Meetings in Toronto, August 15–18, 1983. It is electronically available from StatLib [16], where it is labeled *cars*. The version used in the current work is from the UCI Machine Learning Repository [6], named *Auto+MPG*, in which 8 of the original cars were removed because their mileage values were missing.

**Mine** OBJECTS: 44 coal mines in the Appalachian region of western Virginia. CRITERION: number of fractures in upper seams of coal mines. ATTRIBUTES: inner burden thickness, percent extraction of the lower previously mined seam, lower seam height, duration of operation. SOURCE: The data set is reported by Montgomery, Peck, and Vining [61] and is electronically available from associated R package *mpg* [62] where it is labeled *p13.7*.

**Monet** OBJECTS: 430 sales of paintings by Monet. CRITERION: sale price. ATTRIBUTES: height of the painting, width of the painting, whether the painting is signed, auction house where sale took place. SOURCE: The data set is reported by Greene [37]. It is electronically available from a website maintained by the author [63], where it is labeled *data on sales of Monet paintings*.

**Mortality** OBJECTS: 60 metropolitan areas in the United States. CRITERION: mortality rate. ATTRIBUTES: average annual precipitation, average January temperature, average July temperature, percent population aged 65 or older, average household size, median school years completed by those over 22, percent housing units that are sound and with all facilities, humidity, population density in urbanized areas, percent nonwhite population in urbanized areas, percent employed in white collar occupations, percentage of families with income less than $3000, relative hydrocarbon pollution potential, relative nitric oxides pollution potential, relative sulfur dioxide pollution potential, annual average relative humidity. SOURCE: The data set was assembled by McDonald and Schwing [64]. It is electronically available from StatLib [16], where it is labeled *pollution*.

**Movie** OBJECTS: 62 movies. CRITERION: first-run box office in the United States. ATTRIBUTES: production budget, index of star poser, whether the movie is a sequel, indicator for an action film, indicator for comedy, indicator for animation, indicator for horror, MPAA rating(G, PG, PG13, R), trailer views at traileraddict.com, message board comments at comingsoon.net, attention at fandango.com, percentage of Fandango votes for "can't wait to see". SOURCE: The data set is reported by Greene [37] and is electronically available from a website maintained by the author [63], where it is labeled *movie buzz data*.

**Mussel** OBJECTS: 44 rivers in eastern United States. CRITERION: number of freshwater mussel species. ATTRIBUTES: area of drainage basins, amount of dissolved solids, nitrate concentration, hydronium concentration, number of intervening rivers to four major species-source river systems: Alabama-Coosa, Apalachicola, Savannah, and St. Lawrence. SOURCE: The data are from an article by Sepkoski and Rex [65]. The data set is available electronically from an online repository maintained by Winner [3], where the data set is described as *freshwater mussel species in US Rivers*.

**Obesity** OBJECTS: 136 children. CRITERION: somatotype (a scale of body type, ranging from 1, very thin, to 7, obese). ATTRIBUTES: sex, body measurements at ages 2, 9, and 18: height, weight, leg circumference, strength. SOURCE: The data were collected by Tuddenham and Snyder [66] on children born in Berkeley, California, between January 1928 and June 1929. The data set is reported by Weisberg [22] and is electronically available from associated R package *alr3* where it is labeled *BGSall*.

**Occupation** OBJECTS: 36 occupations. CRITERION: prestige rating of the National Opinion Research Center (NORC). ATTRIBUTES: suicide rate among males aged 20–64, median income, median number of school years completed.. SOURCE: The data set was assembled by Labovitz [67] using data from the U.S. Census of 1950 and prestige rankings obtained by NORC in its 1947 survey. It is reported in a book by Hand et al. [5] with identifying number 490 and label *prestige, income,*

*education, and suicide rates for 36 occupations.* NOTES: For some occupations, median number of school years completed is reported as 16+. These values were treated as 16 in the analysis.

**Oxidant** OBJECTS: 30 summer days in Los Angeles, California. CRITERION: maximum level of an oxidant. ATTRIBUTES: morning averages of four meteorological variables: wind speed, temperature, humidity, insolation. SOURCE: The data set is reported by Rice [68] (pp. 567–570), who notes that the data were collected by the Los Angeles Pollution Control District.

**Ozone** OBJECTS: 13 summers in San Francisco, California. CRITERION: Summer quarter maximum hourly average ozone reading in San Francisco. ATTRIBUTES: Average winter precipitation in the San Francisco Bay area for the preceding two years, summer quarter maximum hourly average ozone reading at San Jose, year of ozone measurement. SOURCE: The data are from a study by Sandberg, Basso, and Okin [69] whose source were measurements provided by the Bay Area Air Pollution Control District. The data set is reported by Weisberg [47] and is electronically available from StatLib [16], in collection *alr*, where it is labeled *alr63*.

**Páramo** OBJECTS: 14 isolated islands of páramo vegetation in the northern Andes. CRITERION: number of bird species. ATTRIBUTES: area, elevation, distance from Ecuador, distance to nearest island. SOURCE: The data set was assembled by Vuilleumier [70] from earlier reports in the literature and his own field studies. It is reported in a book by Hand et al. [5] with identifying number 52 and label *birds in paramo vegetation*.

**Pinot** OBJECTS: 38 samples of Pinot Noir wine. CRITERION: quality. ATTRIBUTES: clarity, aroma, body, flavor, oakiness, region. SOURCE: The data set is reported by Montgomery et al. [61] and is electronically available from associated R package *MPV* [62], where it is labeled *table.b11*.

**Pitcher** OBJECTS: 206 pitchers in North American Major League Baseball. CRITERION: annual salary at the beginning of the 1987 season. ATTRIBUTES: 1986 performance: wins, losses, earned run average, game appearances, innings pitched, games saved; career performance: wins, losses, earned run average, game appearances, innings pitched, games saved; years in major leagues; league at the end of 1986 (American, National); league at the beginning of the 1987 season (American, National). SOURCE: The data set was prepared by the Statistical Graphics Section of the American Statistical Association for the 1988 Annual Statistical Meetings and is available from StatLib [16]. The version used in this work is from Fox [43] and is electronically available from a website maintained by the author [71].

**Plasma** OBJECTS: 315 adults. CRITERION: Plasma retinol level. ATTRIBUTES: age, sex, body mass index, daily caloric intake, daily fat intake, daily fiber intake, daily cholesterol intake, dietary beta-carotene consumed per day, dietary retinol consumed per day, number of alcoholic drinks consumed per week, smoking status (never smoked, former smoker, current smoker), vitamin use (often, used but not often, not used). SOURCE: The data set was made available by Therese Stukel, Dartmouth Hitchcock Medical Center, at StatLib [16], where it is labeled *Plasma_Retinol*. Dr. Stukel notes that a related publication is by Nierenberg, Stukel, Baron, Dain, and Greenberg [72].

**Prefecture** OBJECTS: 45 prefectures in Japan. CRITERION: number of emigrants in the Pacific Northwest in 1911–1912 from the prefecture (per million of the prefecture's population). ATTRIBUTES: percentage of land cultivated by tenant farmlands, change in ratio of tenant farmlands between 1883 and 1907, average area of arable land per farm, number of government contracted laborers sent to Hawaii, whether any of the 18 pioneer Japanese immigrants to the Pacific Northwest were from the prefecture. SOURCE: The data are from an article by Murayama [73]. The data set is available electronically from an online repository maintained by Winner [3], where the data set is described as *Japanese emigration to Pacific Northwest 1880–1915*.

**Rainfall** OBJECTS: 24 days in Coral Gables, Florida. CRITERION: amount of rainfall. ATTRIBUTES: whether the clouds were seeded or not, percent cloud cover, amount of rainfall one hour before seeding, number of days since the first day of the experiment, suitability for seeding, whether the radar echo was moving or stationary. SOURCE: The data were collected by Woodley, Simpson, Biondini, and Berkeley [74] in the summer of 1975. The data set is reported by Weisberg [22] and is available from associated R package *alr3*, where it is labeled *cloud*.

**Reactor** OBJECTS: 32 light water reactors constructed in the United States in the late 1960s and early 1970s. CRITERION: construction cost. ATTRIBUTES: date on which the construction permit

was issued (measured in years since January 1, 1900), time between application for and issue of the construction permit, time between issue of operating license and construction permit, net capacity, whether a prior light water reactor existed at the same site, whether the location is in the north-east region of the United States, Whether a cooling tower is used, whether the nuclear steam supply system was manufactured by Babcock-Wilcox, cumulative number of power plants constructed by each architect-engineer, whether there was a partial turnkey guarantee. SOURCE: The data set is reported by Cox and Snell [75] and Davison [76]. It is electronically available from R package *SMPracticals* [77], where it is labeled *nuclear*.

**Rebellion**   OBJECTS: 32 Romanian counties in 1907. CRITERION: proportion of villages in which rebellious events took place in the Romanian peasant rebellion of 1907, labelled *spread*. ATTRIBUTES: proportion of arable land devoted to wheat, proportion of rural population that is illiterate, strength of middle peasantry (measured by the proportion of land owned in units of 7 to 50 hectares), Gini coefficient of inequality of landownership, population, region (Northern, South Central, Southwest, Eastern). SOURCE: The data set was assembled by Chirot and Ragin [78]. Partial data set is reported by Fox [43] and is electronically available from a website maintained by the author [79].

**Recycle**   OBJECTS: 31 Scottish local authorities. CRITERION: weekly recyclate yield. ATTRIBUTES: weekly recycling capacity, weekly residual capacity, number of principal materials collected, number of extended materials collected, frequency of recycling collection, frequency of residual collection, type of sort (comingled, curbside sort, dual service, single material). SOURCE: The data were obtained by Baird, Curry, and Reid [80] from Scottish local authorities. Partial data set is available electronically from an online repository maintained by Winner [3], where the data set is described as *recycling capacity, items collected and average yield for Scottish local authorities*.

**Salary**   OBJECTS: 52 professors at a Midwestern college in the United States. CRITERION: academic year salary. ATTRIBUTES: sex, rank (assistant professor, associate professor, full professor), number of years in current rank, the highest degree earned (doctorate, masters), number of years since highest degree was earned. SOURCE: The data set is reported by Weisberg [22] and is electronically available from associated R package *alr3* [23].

**Sperm**   OBJECTS: 24 heterosexual couples. CRITERION: mean sperm count per copulation. ATTRIBUTES: age, height, and weight of each of the partners involved, volume of one male teste. SOURCE: The data were collected by Baker and Bellis [81]. The data set is reported by Wood [82] and is electronically available from associated R package *gamair* [83], where it is labeled *sperm.comp2*.

**Tip**   OBJECTS: 244 parties dining in a restaurant. CRITERION: tip rate. ATTRIBUTES: dollar amount of the bill, size of the party, sex of the bill payer, day of the week, time of the day, whether there were smokers in the party. SOURCE: Data were recorded by a food server in a restaurant located in a suburban shopping mall in the United States during an interval of two and a half months in early 1990. The data set is reported in a collection of case studies for business statistics [84]. It is electronically available from R package *reshape* [85].

**Vote**   OBJECTS: 159 counties in Georgia, USA. CRITERION: proportion of uncounted votes in the 2000 presidential election. ATTRIBUTES: type of voting equipment used (optical scan with central count, optical scan with precinct count, punch card, lever, paper), whether the county is in Atlanta, whether the county is urban or rural, proportion of African Americans, economic status (rich, middle, poor). SOURCE: The data set was assembled by Meyer [86]. It is reported by Faraway [41] and is electronically available from associated R package *faraway* [42], where it is labeled *gavote*.

**Waste**   OBJECTS: 20 days of a laboratory experiment. CRITERION: oxygen absorbed by dairy waste kept in suspension in water. ATTRIBUTES: biological oxygen demand, chemical oxygen demand, total Kjedahl nitrogen, total solids, total volatile solids. SOURCE: The data set is reported by Weisberg [22] who notes that the data are from an experiment by Moore [87]. The data set is electronically available from R package *alr3* [23] where it is labeled *dwaste*.

**Wheat**   OBJECTS: 24 samples of ground wheat. CRITERION: protein content measured by the standard Kjeldahl method. ATTRIBUTES: measurements of the reflectance of near-infrared radiation

at six different wavelengths. SOURCE: The data set is presented in an article by Fearn [88]. It is reported in a book by Hand et al. [5] with identifying number 509 and label *measurement of protein content in ground wheat*.

**Women** OBJECTS: 30 famous women. CRITERION: mean attractiveness rating. ATTRIBUTES: mean likability rating, name recognition, whether the woman is American. SOURCE: This prediction problem is from a study by Czerlinski et al. [32]. The data were collected by Henss [60] with the participation of 115 male and 131 female Germans, in ages ranging from 17 to 66 years old.