[Reviews · NeurIPS 2015]

Submitted by Assigned_Reviewer_1

The quality of this work as excellent.

The combination of a solid theoretical analysis and unusually good experimental investigation round the paper out nicely.

The clarity and quality of the presentation is high, and moves the reader through the ideas well.

The work isn't stunningly original in its approach, but but the through and methodical investigation lead to several novel results.

The significance of this work is substantial within the area of heuristic decision making.

My one concern is that the population of those who should be most impacted by this work is not particularly high at NIPS, but the technical intricacy of the work might make NIPS one of the better venues for presentation.

If the paper isn't accepted to NIPS, I would suggest the authors consider a venue like the Journal of Experimental Analysis of Behavior, or other space amenable to mathematical treatments of decision making.
Summary: A surprising paper on the effectiveness and learning dynamics of simple decision making heuristics.

Several surprising results, enjoyable reading, and with better empirical validation than most work in this area.

Submitted by Assigned_Reviewer_2

First provide a summary of the paper, and then address the following criteria: Quality, clarity, originality and significance.

This paper reviews and evaluates on 63 diverse datasets the performance of simple decision heuristics for the 'comparison problem':

choosing the better of two objects from multiple ordinal cues (attributes).

To be informative, a cue has to be different when the objects have different values.

Single-cue:

Select the most informative cue from the training sample and discard the rest.

Ties are broken randomly.

Take-the-best:

Use the most informative cue whose value differs on the two objects being compared.

Tallying:

Each cue votes for one or the other object if the cue differs for the objects, and otherwise abstains.

The decision is based upon the number of votes.

Ties are broken randomly.

These simple decision heuristics were compared against a weighted sum (logistic regression with binarized cues) and a decision tree classifier.

For large training data sets, performance of the take-the-best approach was surprisingly close to logistic regression and decision trees, while tallying did not perform well.

Performance as a function of training set size was also examined.

For small training sets, tallying does best and logistic regression does worst.

Quality:

The study is competently done.

Clarity:

The paper is quite clear and well-written.

However fonts on most figures should be larger.

Originality:

Not my domain, but it seems like this may be the first paper to empirically evaluate these heuristics.

Significance:

The heuristics considered here are technically simple.

However, many complex artificial systems do

use simple heuristics at various stages, and having some empirical data on the general reliability of these heuristics is useful.

Particularly relevant here is the variation with training set size, which has a massive impact on which system performs best.

There is some brief discussion relating this to human decision-making, but since this paper does not evaluate human decision making, or seriously review any of the literature in this area, the connection is a bit weak.
Summary: Competent study on the empirical effectiveness of simple heuristics for comparison problems, with particular focus on the dependency on the size of the training set.

Seems reasonably novel and of some practical value.

Submitted by Assigned_Reviewer_3

The authors evaluate a number of decision rules from the ABC research group. They first analyze, through simulations, that the "cue direction" and "learning to order cue" heuristics can provide accurate performance with even just a few samples. Next, they evaluate a number of heuristics on 63 data sets and demonstrate that some of them achieve performance close to an additive logistic regression and CART, with default parameter settings.

This is a well-written paper containing two substantial results that I believe will be of interest to a broad community at NIPS. I have a few main criticisms regarding the submission (in order of appearance). The first result they have (few samples suffice for some heuristics) is very similar in nature to a previous result by Vul, Goodman, Griffiths and Tenenbaum (2009; 2014). Although the results are not identical, I did not find the simulations terribly surprising or impressive given this previous work. Also, I am confused why the simulations discuss cue direction and informative sampling when neither are discussed nor used later in the paper. The other main criticism is concerned with the models used as comparisons to evaluate the performance of the heuristics on the 63 data sets. A simple logistic regression where each cue is an independent input is not that powerful of a statistical model. What if pairs and triplets are included as well? What about support vector machines/kernel methods on these data sets? I know this increases the computational complexity of the comparison models, but if the argument is that heuristics are close to the best models, than the best models should really be powerful. I was more impressed with CART, but the fact that the default settings were used with no exploration made me concerned that CART is underperforming because there is no tuning whatsoever to each data set. If the best parameters are chosen for CART on each data set, how well would it do? What are the best parameters for CART that do the best across all data sets?

Minor concerns: Line 131: it might be worth mentioning that e1 is simply the CDF of a binomial distribution. Figure 1: The text is too small in many places and I found the subplots difficult to interpret. Also, the scale of the colormap for the bottom left subplot is different than the other subplots. Line 201-202: "..., but the in the majority" -> "..., but in the majority" Line 219: Wouldn't it be more appropriate to use ordinal logistic regression, (polyr in the MASS library of R), rather than turn an ordinal factor into a nominal factor? Line 258-259: "with upto 3000 repetitions." -> "with up to 3000 repetitions."

Vul, E., Goodman, N., Griffiths, T., & Tenenbaum, J. (2014). One and Done? Optimal Decisions from Very Few Samples. Cognitive Science, 38, 599-637.

Vul, E., Goodman, N., Griffiths, T., & Tenenbaum, J. (2009). One and done? Optimal decisions from few samples. In N. Taatgen, H. van Rijn, L. Schomaker, & J. Nerbonne (Eds.), Proceedings of the 31st annual conference of the cognitive science society. Amsterdam, the Netherlands.

Edits based on author feedback: I'm glad to hear the initial review was helpful. The paper is a really great contribution and your feedback increases my confidence in that assessment. I apologize regarding the confusion with "pairs and triplets". What I mean is including interaction terms between your predictors in the logistic regression. Perhaps you did that and I misunderstood what you did in the manuscript? If so, it's probably worth revising the section on logistic regression a bit to make this clear.
Summary: A well-written paper describing an impressive result of how accurate simple rules can be. However, the comparison models as powerful as they could be and their first result is very similar to a previously published result.

Author Feedback
Author rebuttal: We thank all reviewers for their time and thoughtful comments.

**Reviewers 2 & 4**
There is substantial work in the psychological literature on take-the-best and tallying as models of human decision making. We would be happy to discuss this literature in more detail. Our aim is to complement earlier work with a computational analysis. Our results are important both for designing artificial agents and for further developing models of human decision making.

**Reviewer 4**
We describe the data sets in detail in the supplementary material. NIPS archives the supplementary material so this information will be available to the readers.

**Reviewer 6**
The reviewer lists two main weaknesses. We respectfully disagree on both points.

The first is lack of novelty because all methods "have been used in previous works". Our paper is the first to evaluate single-cue; it provides the first theoretical and empirical [1] results on learning dynamics; it is the most extensive empirical evaluation of these heuristics to date [2]. Our conclusions are novel.

The second is that experimental results show improvement when training size is smaller than 40. This is not exactly correct: learning may progress beyond 40 samples (example: bodyfat, Fig 4). But it is true that learning usually happens quickly. We believe this is a strength: learning with scarce data is important in many real-world problems.

**Reviewer 9**
We should clarify that CART performs cross-validated cost-complexity pruning (this is part of the standard CART implementation). The parameters minsplit, cp, and minbucket control only the size of the initial tree to be used for pruning. The actual tree obtained after pruning is in fact tuned to the individual data set and the given training set size. (Apologies, we should have stated this explicitly in the paper.)

We repeated the analysis with SVMs (linear kernel, cross-validated grid search on the hyper parameters). The results are not qualitatively very different than those presented in the paper. Performance on individual data sets varied: there are some data sets where TTB and Tallying are consistently better than SVM; there are also data sets where SVM dominates the heuristics over the entire range of sample sizes. When we examine mean performance in 63 data sets, we see that (1) tallying is better than SVM when training sample is small (7 or less) but lags behind as sample size grows, (2) take-the best performs as well as or slightly worse than SVM for most sample sizes. We would be happy to include this additional analysis in the paper.

There is an interesting connection between our work and the work by Vul et al. The papers share a similar formal analysis but the research questions are different. Vul et al. investigate how many samples are required from the posterior distribution in order to approximate the Bayesian solution. In contrast, we examine the learning process of specific decision heuristics (which are in fact alternatives to Bayesian cognitive models). Decision problems/methods considered in our work differ considerably from those by Vul et al. (e.g., their "samples" are from the posterior distribution in hypothesis space following new real-world experience; our "samples" are the real-world experiences themselves.)

We intended the formal results in Section 3 and the empirical results in Section 4 to be complementary. In Section 3, we isolated two building blocks of heuristics (determining cue direction and determining cue order) and analytically examined their learning rates. Defining an "informative sample" was necessary for doing that. We will strengthen the connection between the two sections.

"What if pairs and triplets are included as well?" We do not follow what the reviewer means here. We applied logistic regression to the comparison problem in the standard way (e.g., Martignon & Hoffrage, 2002, Theory and Decision). We would be happy to do additional analysis if the reviewer would provide a short explanation.

Footnotes:
[1] When examining learning dynamics, the most appropriate unit of learning is independent, identically-distributed paired comparisons. This is what we examine in this paper. No earlier work, empirical or theoretical, have done this. A few earlier papers show empirical performance of heuristics as a function of number of training objects (references 4-6 in the paper). These results are very limited in scope: they used few data sets, converted all predictors to binary, etc. More importantly, they do not reveal the effect of each new iid training sample. For example, 5 objects yield 10 distinct paired comparisons; a sixth object yields 5 additional paired comparisons (which are not independent from the earlier sample of 10).

[2] We use 63 data sets while the largest earlier empirical work used 20 data sets. Furthermore, this earlier work dichotomized all predictor variables around the median, converting all variables to binary.